# Bilinguals’ Working Memory (WM) Advantage and Their Dual Language Practices

**DOI:** 10.3390/brainsci7070086

**Published:** 2017-07-18

**Authors:** Eunju Yang

**Affiliations:** Board of education in Incheon, Incheon 21554, South Korea; eunjuyan@buffalo.edu

**Keywords:** bilinguals, working memory, mixed methods, language practices

## Abstract

The present study investigates a possible working memory (WM) difference between monolingual and bilingual groups and explores the relationship between their WM advantage and language practices. A mixed methods design was employed for the study. To measure participants’ WM, auditory and visual digit span tasks were conducted on the different language groups: 20 Korean near-monolinguals, and 40 Korean–English bilinguals with two different levels of second language (L2) proficiency. Bilinguals’ daily language practices were explored through semi-structured interviews with eight bilinguals. The convergence of the findings from both tests and interview data suggests that knowing two languages does not guarantee bilingual WM advantages over monolinguals, but the advantage might be linked to bilinguals’ unique L2 use environment where they need to hold incoming L2 information while decoding it.

## 1. Introduction

Cognitive differences between monolinguals and bilinguals have been widely studied over the last few decades. Studies show evidence in favor of bilinguals’ cognitive benefits in various areas such as problem solving [1,2], metacognitive awareness [3,4], divergent thinking [5,6] and attention control [7]. The main reason suggested for bilinguals’ advantage is their need to process and manage the two languages, which are simultaneously activated whenever one of the languages is used [8,9,10,11]. This simultaneous activation requires a higher working memory (WM) capacity. Studies suggest that bilinguals’ two activated languages impose a heavier cognitive load so that their WM might be, to a certain degree, influenced by the dual language practices [12]. However, studies on bilinguals’ WM have provided conflicting results so it remains inconclusive whether bilingual WM advantages exist or not. The present study tests WM differences among three language groups with different second language (L2) proficiency levels to find out whether bilingualism grants any WM advantage and whether the advantage relates to bilinguals’ L2 proficiency. It also uses interview data to explore how bilingualism influences memory use since research suggests that cognition changes might depend on bilinguals’ dual language practices. 

## 2. Working Memory and Language Practices

### 2.1. Working Memory in Bilingualism

Working memory refers to a function of the brain that holds, manipulates, and processes temporary information needed to accomplish various tasks at any given moment [13,14]. Because of the nature of constant information processing of WM, it has been suggested that WM plays a significant role in diverse cognitive processes such as language comprehension, planning, reasoning, and problem-solving where holding incoming information in mind is required [15,16,17,18,19].

According to Baddeley and Hitch’s [20] widely used WM model, there are three components of WM: the central executive, the phonological loop, and the visuospatial sketchpad subsystems. The central executive is an attentional controller system, which is assisted by two independent subsystems: the phonological loop and the visuospatial sketchpad. This system makes decisions on how to use incoming information in the two subsystems and allows one to hold or drain information from those subsystems when they are overloaded [21]. The visuospatial sketchpad subsystem deals with visual and spatial information, while the phonological loop subsystem processes sound or phonological information. These three components collaboratively work together to process information for ongoing tasks. 

WM is a component of executive functions (EFs). EFs are cognitive processes that regulate goal-directed human behavior as well as control human thoughts and responses in an automatic or established manner [22,23]. It has been suggested that EFs consist of three components: WM, inhibition, and shifting [24,25]. For decades, many studies have been done on the components of EFs, and have shown that bilinguals have advantages in inhibition and cognitive flexibility over monolinguals [26,27,28,29,30,31]. These studies have suggested that bilinguals’ practices in controlling two activated languages develop their ability to focus on one thing over distractions and that their practices in switching between languages develop their cognitive flexibility [24,25]. Currently, Miyake and Friedman proposed a view on EFs that explains that the components of EFs are highly intercorrelated and function as one unit even though each component is separable [32]. According to this view, it is highly probable that bilingualism has an influence on WM as it does in two other components of EFs. 

Furthermore, it is believed that bilinguals’ dual language processing imposes a heavier cognitive load on their WM. This heavier cognitive load leads to two possible hypotheses about bilinguals’ WM. The first is that bilinguals might have a disadvantage in terms of their WM function because of their heavy language load from two activated languages [12]. The alternative is that they might have an advantage in WM function because of their possible development of an efficient mechanism for managing two simultaneously activated languages, as shown in their inhibitory control and cognitive flexibility [7,32]. 

### 2.2. Findings in Bilingual WM Studies 

While some studies have shown that bilinguals have cognitive advantages in inhibitory control and cognitive flexibility [26,27,28,29,30,31], research on bilinguals’ WM has been conducted to provide sufficient evidence to support either of these hypotheses. 

Some studies measured bilinguals’ WM as a part of the executive control system. For example, Bialystok et al. [7] used the Simon task to examine the age effect on bilinguals’ EF advantage. In the study, participants were required to react to stimuli with a different level of WM manipulation and the reaction times to the stimuli were measured. The bilingual group performed similarly to the monolingual group in the task with light WM manipulation, but bilinguals outperformed monolinguals with the heavy WM manipulation. This indicates that bilinguals have WM advantages over monolinguals as well as that WM and executive control are somehow related to each other. A similar study was conducted on monolingual and bilingual children [33]. In the study, bilingual children outperformed monolinguals and maintained their outperformance in all tasks with heavier memory load tasks. The result suggested that bilingual children have more efficient information management skills than monolingual children. 

Studies were also conducted with simple WM tests and showed conflicting findings. For example, Blom et al. [34] investigated visuospatial and verbal WM differences between Turkish–Dutch bilingual children and Dutch monolingual children. Their findings support bilingual advantages in both visuospatial and verbal WM. Meanwhile, studies also showed evidence that failed to support bilingual advantages in WM. In verbal WM tasks, the fluent bilingual children did not perform better than those who were not fluent [35]. There was no difference in visual and verbal WM between cross-language monolingual (French and English monolinguals) and bilingual groups [36], and there was no visuospatial WM advantage between bilingual children over monolingual children [37]. 

These inconsistent findings do not offer a clear picture of bilinguals’ WM advantages, and thus require further investigation. 

### 2.3. Degree of Bilingualism 

Most of the studies mentioned have been conducted on children who learned L1 and L2 simultaneously or immigrants who moved to their L2 country early in their lives and became balanced bilinguals. Not much research has been conducted on sequential bilinguals (who learn their L1 first, then L2 later) with different levels of language proficiency. If bilingualism influences bilinguals’ cognitive processes, it is possible that the cognitive impact gradually appears as bilinguals gain higher degrees of bilingualism. According to Cummins’s threshold hypothesis [38,39], a threshold level in language proficiency in L1 and L2 should be attained in order to demonstrate the cognitive advantages of bilingualism. However, there has been no clear evidence of the relationship between bilinguals’ cognitive advantage and language proficiency. Therefore, it is important to examine bilinguals’ WM in relation to the degree of bilingualism.

### 2.4. Cognitive Advantages and Language Practices 

As discussed, studies suggest that bilinguals’ use of two languages serves as mental training and enhances bilinguals’ cognitive ability [7,26,30]. Bilinguals develop the ability in inhibitory control while managing two activated languages and their cognitive flexibility through switching between two languages [7,24,25,26,30]. Recently, studies have further suggested that bilinguals’ cognitive advantage is not only affected by knowing two languages but also by how they use as well as experience their languages in their lives [40,41]. Bilingual groups have various language practices depending on their L1, L2, and cultures [42,43]. As each language group displays its own distinctive way to practice a second language (L2), the pattern of language uses might have a unique impact on their cognitive change. Not many studies have closely explored bilinguals’ language practices in relation to their cognitive changes. Therefore, it is important to consider bilinguals’ language practice tendencies on examining their cognitive advantages. 

### 2.5. The Purpose of the Study 

Considering all these factors, the purpose of the present study is threefold. First, it aims to examine possible cognitive differences in WM between monolingual and bilingual groups, and it further examines whether WM differences exist between two bilingual groups with different language proficiency. A young adult Korean near-monolingual group and two young adult Korean–English bilingual groups with different levels of language proficiency were chosen for the study. Second, the study aims to investigate how bilingualism influences memory use by exploring how bilinguals in each group practiced their dual languages in everyday life. Third, the study aims to explore any possible relationship between bilinguals’ WM differences and their daily language practices. In order to accomplish these purposes, the study employs the parallel mixed-method design [44]. A possible difference in WM among three language groups was measured with the Digit Span Tasks (DST), which specifically measure visual and spatial memory maintenance and manipulation ability. In order to describe the daily language experiences of bilinguals in detail, semi-structured interviews were employed. Participants’ demographics were strictly controlled to eliminate any compound effects from their linguistic and cultural backgrounds. In addition, the bilinguals chosen for this study were selected based on their language proficiency, immersion language exposure, and daily language use since research suggests that bilinguals’ L2 use is positively correlated to their L2 proficiency [38,45] and that bilinguals’ cognitive advantages result from their continuous use of two languages. 

## 3. Materials and Methods 

### 3.1. Participants

The sample consists of three groups of 20 adults in their 20s and 30s: a Korean near-monolingual group, a Korean–English bilingual group with intermediate L2 proficiency, and a Korean–English bilingual group with high L2 proficiency. In the recruiting stage, potential participants’ gender, age, field of study, and socioeconomic status (SES) were checked through a pre-designed questionnaire, and these factors were carefully controlled in order to minimize their influence on the results of the study. The near-monolingual speakers were South Korean college students who speak only Korean and who have no experience living in countries other than South Korea. The study recruited “near-monolinguals” because Koreans are required to learn English in the school system so that they are exposed to English. However, monolingual participants with very limited English proficiency were selected. All bilingual participants were university students in the Buffalo, NY area, and were sequential bilinguals who acquired their L2 after their L1. They started learning the L2 in the school system in South Korea around the age of 10 and started improving their L2 in the English immersion environment when they moved to the United States. They arrived in the United States after the age of 12. 

Historically, in linguistic studies, the degree of bilingualism has been decided by three factors: the amount of language use, age of acquisition (AoA), and L2 proficiency [46,47,48,49]. The present study considered the amount of language use and L2 proficiency to distinguish between different bilingual groups while controlling AoA by recruiting only sequential bilinguals. Thus, the following criteria were set for selecting bilinguals with intermediate L2 proficiency: equivalent use of L1 and L2, three to five years of living in the United States, and L2 proficiency (iBT Test of English as a Foreign Language (TOEFL) scores). The range of iBT TOEFL score of bilingual participants with intermediate L2 proficiency was set from 65 to 90 based on the interpretation of TOEFL scores in Table 1. The score ranges for reading, listening, speaking, and writing are set at 15–21, 15–21, 18–25, and 17–23, respectively. The mean TOEFL score of the intermediate bilingual group of the study was 83.7.

The bilingual participants with high L2 proficiency were also selected based on three factors: equivalent use of L1 and L2, more than five years of exposure to the L2 environment, and high L2 proficiency (iBT TOEFL score). The range of iBT TOEFL scores for bilinguals with high L2 proficiency were set between the score of 94 and 120, and the minimum iBT TOEFL scores for reading, listening, speaking, and writing were set at 22, 22, 26, and 24, respectively. The mean TOEFL score of this group was 101.4. 

Each group had four students who study economics, two natural science, four nursing (including pharmacy and physical therapy), three education, two social studies, two international studies, and three engineering. For this study, students who had lived in the USA for fewer than three years and whose L1 and L2 use was not balanced were excluded since the study assumed that bilinguals’ daily language use might play a significant role in their cognitive changes. The description of all participants is in Table 2. All subjects gave their informed consent for inclusion before they participated in the study. The protocol of the study was approved by the Ethics Committee of SUNY University at Buffalo Social and Behavioral Sciences IRB (IRB00003128). 

### 3.2. Questionnaire

Two separate questionnaires were used to select participants: one was for Korean near- monolinguals, and the other was for both Korean–English bilinguals. The questionnaire for Korean monolinguals included their gender, age, field of study, language background, education background, and socioeconomic status (SES), as well as their experience with languages other than Korean in order to select monolinguals who either did not have or had minimum exposure to other languages. The questionnaire for bilinguals included all the same questions as the questionnaire for near-monolinguals plus additional questions about the following factors: their average weekly hours of each language use, the number of years living in the USA, and their English proficiency level based on the their TOEFL scores. All participants had official valid language test scores when the study was conducted. The information from the questionnaire is presented in Table 2. 

### 3.3. IQ and Age 

A review of previous literature yielded evidence that there is a relationship between WM and general fluid intelligence [51,52]. Even though there is still an argument that general intelligence correlates with WM, the present study conducted intelligence tests with all participants in order to minimize the compound effect of general intelligence on WM tasks. The Cattell Culture Fair Intelligence Test [53] was used for the study as it is a nonverbal test without linguistic and cultural influence. In addition, participants’ average age was also checked since age is related to individuals’ performance on IQ tests [54]. 

Two separate one-way ANOVA tests were used to analyze the group differences in age and IQ. In order to control the Type I error rate, Bonferroni correction was used with a modified alpha level at *p* = 0.025. The descriptive statistics associated with age and IQ across the three language groups are presented in Table 3. The ANOVA results showed that there was no statistical difference in means of age, *F* (2, 57) = 0.034, *p* > 0.025, as well as in means of IQ scores, *F* (2, 57) = 0.498, *p* > 0.025, among the three different language groups. This indicates that the age and IQ of the three language groups were similar to each other; therefore, it was unlikely that differences in performance on the cognitive tests would be affected by their age or intelligence. 

### 3.4. Digit Span Tasks (Auditory and Visual)

This study employed the computerized Digit Span Tasks (DSTs) which measure the visual and auditory digit span. To control any language effects on the DSTs, numeric digits were given as stimuli for the visual DST and Korean was chosen as auditory stimuli for the auditory DSTs since all the participants’ first language is Korean. 

The DSTs is one of the oldest neuropsychological tests used for measuring short-term memory [55]. The tasks in the present study used strings of one digit numbers presented in random order to test auditory and visual digit spans. The study added the backward DST in order to provide a better measure of WM as the forward DST tests short-term memory. In the test trials for the auditory DST, participants were asked to listen to a sequence of auditory digits and hold the sequential information of numbers. The auditory digit started with three digits, and each digit was presented for one second. After hearing the presented digits, participants were asked to type the numbers into a textbox presented in the middle of the computer screen. When participants typed the sequential numbers in the correct order, they moved into the next trial where sequential numbers with one more digit was presented. If a participant made a mistake, a second chance was given. A participant who failed to get the correct answer two times in a row was moved back to the previous trial that had one digit fewer. The trials were designed to continue for up to 14 trials, which is adapted from the experiment of Woods, et al. [56]. The computerized DST recorded the following information: Two-error maximum length (TE-ML), two-error total trials (TE-TT), maximal length (ML), and mean digit span (MS). TE-ML is the maximum digit span that a participant reaches before making two consecutive errors. This is a traditional measure of digit span, which is assumed as true maximum digit length. TE-TT records the total number of trials that a participant gets before reaching TE-ML. ML is the maximal digit span that a participant recalled correctly during all 14 trials while MS records the mean digit span that a participant is expected to get correct 50% of all times based on overall performance during 14 trials. 

Both the auditory and visual DSTs had two sets of tests: the forward DST measures memory maintenance and the backward DST measures memory manipulation capacity. In the backward DST, all of the procedures were applied in the same manner as in the forward DST, except the fact that participants were asked to recall the digits in the reverse order of the presented numbers. All the rules in the auditory DST apply for the visual DST in the same manner, but the digits were presented visually in numeric numbers in the middle of the screen. The visual tasks also included the backward DST. A total of eight separate scores were recorded for each auditory DST and visual DST. In order to control the possible study effect across the auditory and visual DST, half of the participants took the auditory DST first, while the rest of them took the visual DST first. 

### 3.5. Semi-Structured Interviews 

Semi-structured interviews were conducted to explore how bilinguals practice their dual languages in their daily lives. All bilingual participants who volunteered for participating in cognitive tests were asked to indicate their willingness to participate in interviews on the consent form prior to their cognitive tests. Participants who were willing to be interviewed were asked questions about their language experiences when they first came for the cognitive tests. The participants’ willingness to participate in interviews and to share their stories was considered as a primary factor for selection. Other factors such as age, gender, and educational background were also considered when selecting representative interview participants from each language group. Four participants from each bilingual group were recruited for interviews, and their information is provided in Table 4. All participants were de-identified using alphabetical pseudonyms.

The interviews were conducted in three one-hour sessions: the first and second sessions were individual sessions with each participant, and the third session was a group session with all four participants of each group. The first interview started with the pre-developed interview protocol. This protocol was designed to obtain a rich description on their daily language practices and their cognitive efforts while learning their L2. The second interview protocol was designed based on information shared in the first interview. The third, group interview was done with questions from both the first and second interview in order to determine to what extent the interviewees shared similar experiences in their language use and to see how the group reacted to the questions together [57,58,59]. The interviews addressed the participants’ daily language usage, dual-language practices, language learning strategies, language management, and experience in learning and using two languages.

## 4. Analysis

A multivariate analysis of variance (MANOVA) was used for analyzing the scores from the DSTs as it has more than two dependent variables and those variables in each test measure the same construct of each component [60]. The MANOVA protects against inflating the Type I error rate from the series of the post-hoc comparisons followed by the main analysis with a significant difference [61]. 

For interview data, 18 interview recordings were transcribed and then analyzed using thematic analysis [62]. First, the interview data were carefully read and reread in order to identify meaningful units of text related to the purpose of the study. In this stage, each turn of talk reflecting bilinguals’ language usage, practices, and mental efforts in learning language was identified and extracted from the original transcriptions for the further analysis. Second, codes were assigned to each sentence in extracted texts and nearby sentences carrying the same codes were combined. Third, codes were examined in order to find similarities or relationships between codes. When codes carried similar issues or were related to each other, they were grouped together in a category, which reflected all the codes under the category. The second coder was invited from the first stage of coding and the codes developed separately by the two coders were compared at each stage. When there were disagreements between the two coders, the text of disagreement was excluded from the data set. The analysis developed six categories that were grouped into two themes. The themes and categories are reported in Table 5.

## 5. Results

### 5.1. Results from Digit Span Tasks

The auditory and visual Digit Span Tasks (DSTs) yielded four separate scores and were recorded for each DST session: Two-Error Maximum Length (TE-ML), Two-Error Total Trials (TE-TT), Maximal Length (ML), and Mean Digit Span (MS). In total, 16 variables were measured. Prior to conducting analysis, Pearson correlations between dependent variables were conducted in order to check one of the MANOVA assumption: the dependent variables would have moderate correlation range with each other [63] (i.e., 0.02–0.06). The correlations between variables are presented in Table 6.

There were strong (>0.70) correlations between pairs of variables, for example, between TE-ML and TE-TT, between TE-ML and ML, and between TE-ML and MS. This strong correlation appeared in every forward/backward auditory/visual DST section. It was assumed that those pairs of variables which show high correlations with each other were measuring the same construct (for instance, participants who scored highly in Two-Error Maximum Length had high Two-Error Total Trials since they needed to complete more trials in order to get a higher digit span; participants with high scores in Two-Error Maximum Length generally reach higher in Maximal Length). Therefore, the study decided to keep the one variable that is the most informative among the four. TE-ML was selected for further analysis as it had information on the actual maximum digit span and also showed moderate correlation with each of the other three variables. Thus, the following four variables were included for main MANOVA analysis: forward TE-ML and backward TE-ML for each auditory and visual DST. Correlations between these selected variables were checked and presented in Table 7. A moderate range of correlations between most of the dependent variables was observed, suggesting the appropriateness of a MANOVA analysis.

In addition, the homogeneity of covariances in dependent variables was tested through Box’s Test of Equality of Covariance Matrices. The Box’s M value of 22.137 was associated with a *p* value of 0.466, which indicates that there was no significant difference between the covariances of dependent variables. Therefore, the MANOVA was appropriate for the dataset. 

A one-way MANOVA was conducted to test whether there was any mean difference between DST scores of the three different language groups (near-monolinguals, intermediate bilinguals and high bilinguals). A statistically significant *F* value was obtained from the MANOVA tests, Wilks’ Lambda = 0.728, *F* (8, 108), *p* < 0.05, showing that there was difference in DST scores among the three language groups. The multivariate effect size was estimated at 0.147, which implies that 14.7% of the variance in the canonically derived dependent variable was accounted for by language groups.

In order to determine where the language group differences lay, a series of follow-up ANOVAs were chosen as a post hoc procedure. Prior to conducting ANOVAs, the homogeneity of variance assumption was tested for all four DSTs. Based on a series of Levene’s *F* tests as seen in Table 8, it was considered that the homogeneity of variance assumption was satisfied, therefore, the ANOVAs were appropriate in this case. 

A series of one-way ANOVAs was conducted on each of the four variables. As seen in Table 8, the results of ANOVAs on visual DSTs were not statistically significant (*p* > 0.05), suggesting that there was no language group difference in both the visual forward and the backward DSTs. However, it should be pointed out that the ANOVA on the visual forward TE-ML was close to a 0.05 *p* value. Therefore, it would be meaningful to look at each group’s mean difference in the visual forward TE-ML. Meanwhile, ANOVAs on both the auditory forward and backward DSTs appeared to be statistically significant, with large effect sizes (Partial *η*^2^): 0.134 for the auditory forward TE-ML and 0.218 for the auditory backward TE-ML. This suggests that there were mean differences in both the visual forward and the backward DSTs across the language groups. 

In order to examine individual mean difference comparisons across three language groups and all four variables, a series of post hoc analysis (Fisher’s LSD) were conducted as seen in Table 9, and the mean TE-ML scores for group comparisons are presented in Figure 1. 

The results revealed that the intermediate bilingual group showed significantly higher TE-ML scores than the other language groups in the visual forward TE-ML and the auditory forward/backward TE-MLs. When the effect sizes as estimated by Cohen’s *d* were examined, the medium effect sizes were observed in the visual forward TE-ML with Cohen’s *d* value at 0.62 (between the near-monolingual and intermediate bilingual groups) and at 0.73 (between the intermediate bilingual and high bilingual groups). The large effect sizes were observed with both the auditory forward and the backward TE-MLs with Cohen’s *d* value ranging 0.83 to 1.01 [64].

Overall, the findings showed that intermediate bilinguals, in general, performed better in the visual forward DST and the auditory forward/backward DSTs. This suggests that the intermediate bilinguals have higher visual/auditory memory maintenance as well as higher auditory manipulation capacity than the two other language groups in the study. It is noticeable that the higher effect sizes were associated with auditory memory, suggesting that auditory WM differentiates the intermediate bilingual group the most from other groups. Meanwhile, statistically significant mean differences were found only between intermediate bilinguals and the other groups, not between the near-monolingual and high bilingual groups. 

In sum, the results of DSTs showed that the three language groups differed in their visual short-term memory and auditory WM capacity. The intermediate bilinguals performed better than the two other language groups, while the high bilingual group did not differ from the near-monolingual group. This indicates that bilingualism does not guarantee a WM advantage and that bilingualism might have a different effect on WM depending on bilinguals’ L2 language proficiency. 

### 5.2. Results from Semi-Structured Interviews

WM is related to the capacity to hold information in mind. The interview data showed that bilinguals in the study expend cognitive efforts to remember incoming L2 input by developing strategies when using their L2. As seen in Table 4, two themes, *Strategies in using 2nd language* and *Efforts is continuous*, and three categories under each theme emerged from the analysis of data. 

#### 5.2.1. Strategies in Using a Second Language

Bilinguals reported that they had developed language use strategies to manage the two languages. Their strategies and the reason for using them are well described under three categories: difference between L1 and L2, monitoring languages, and holding information. These three categories are explained together with excerpts. 

Difference between L1 and L2. All bilingual participants were aware of the language differences between Korean and English since they started learning their L2 with their fully developed L1. The language difference was one of the reasons why they felt difficulties in learning their L2. However, the difficulties led them to develop their own strategies to overcome them. For instance, participant B commented on her experiences on language differences from her early stage of learning English:
*English and Korean have different sentence structures: in English, verbs follow subjects, but in Korean, verbs are at the end of the sentences. It was very confusing when I started learning English. It was like I needed to remember the whole sentence and reorganize it to fully understand what I hear*.She said that Korean and English have different sentence structures. This made her remember the sentence that she heard, then she re-organized it in order to understand. In her comment, “*Reorganize*” means that she rearranged English words in the sentence into the Korean sentence structure to interpret the meaning of the sentence. When she was asked whether she meant “*reorganize*” by “*translating L2 into L1*”, she said: “*not really; I need to know where the verb is.*” Her comment implied that, even though she was not translating, she put English words into the frame of her native language structure. Considering the different word order between English (Subject + Verb + Object) and Korean (Subject + Object + Verb), her re-organizing indicated that, at least in her early stage of L2 learning, she relied on her L1 system, to some degree, to understand her L2 input. Based on her comment, it can be assumed that her L2 system had not reached a level at which she was able to understand as she was hearing the L2 input, so that she needed to use her L1 system to understand. All interview participants reported this practice in their early L2 learning stage.Monitoring languages. Most of the bilingual groups paid extra attention to their L2 output to monitor it, even if the degree of attention and the language features they focused on varied between the participants (this is discussed in detail under the theme ‘Effort is continuous’). The language features to which they attended included pronunciation of words, usages of words, and language structures. Participant G described how he was attentive to the L2 grammar when he communicated with people in English:
*I think I pay attention to the language forms when I listen to English, cuz most of the time, I can tell “oh that person didn’t put 3rd person singular”, “Oh, my tense is wrong”, but I don’t do those kind of things when I hear Korean. Korean comes so naturally I do not even need to think*.He was attentive to his L2 form and his attention enabled him to pinpoint his L2 errors in his speech. However, he did not intentionally focus on his L2 output. For example, he said he “*thinks*” he “*pays*” attention to the language form because he was able to notice his own errors, while he was not able to notice any errors when he used Korean; as he stated: “*I do not even need to think*.” G’s comment also implied that he had a tendency to check the L2 forms and his tendency became automatic and unnoticed unless he found himself producing errors. Most bilingual participants across the language groups reported that they had a tendency to monitor their language in attempts to produce proper and correct language. Participant H shared an example of his monitoring and his reason to monitor his L2:
*Even though I think ahead, sometimes I make mistakes, but I always think. I need to say proper words and correct sentences so people will understand me. So that is why I think, especially when I say longer sentences, I think carefully, I focus on what I need to say…I kinda remember words and sentences that I use, you know, cuz I can tell whether my English is correct or not as I speak*.He said he wanted to produce “*proper words*” and “*sentences*” so that people could easily comprehend his English. H monitored his language to produce “*correct*” English, and this monitoring was done carefully with focus; as he said, “*I focus on what I need to say.*” This can be interpreted to mean that he pays extra attention to what he wants to focus on when he uses English. He also said that he “always thinks” about what he was going to say ahead of time. This implies not only that he monitors his L2 intentionally every time he uses it but also that this monitoring has become habitual for him since he does it whenever he uses his L2 to avoid making mistakes. His comment, “*I can tell whether my English is correct or not as I speak*,” indicates that his monitoring has become an automatic process so that he can recognize his mistakes whenever they are committed. Holding information. Bilingual participants across the groups frequently mentioned that they tried to remember what they heard when they conversed in English. However, there were differences between participants as to how they remembered information and why. The pattern that the intermediate bilinguals showed was mostly remembering sentences. Participant E is a representative example of bilinguals’ sentence remembering strategy:
*When I first started learning English, I needed to remember what I heard so I could understand, I mean, I needed to play what I heard in my mind if I didn’t understand what I heard back then. It was hard for me to understand as I heard, you know what I mean, so I could keep thinking [about] the sentence so I could get the meaning. Sometimes, when I heard a very difficult sentence, I needed to translate sentences into Korean to have full understanding, so I needed to remember what I heard*.Participant E tried to remember what she had heard in order to understand the conversation correctly. She reported that when she first started learning English, she tried to hold sentences in her mind so she could replay them when she could not understand the meaning of sentences as she heard them. It seemed that this replaying activity saved some time for her to decode the meaning of the sentences. Participants often reported that they also needed to replay the input in order to translate it into their L1, especially in the early stage of L2 acquisition; as Participant E said, “*I needed to translate.*” Translation seems to help them understand the conversation in the early L2 learning stage, when their emerging L2 skills challenged them to understand the ongoing L2 conversation. In other words, their lack of L2 proficiency caused them to develop strategies to replay their L2 input and translate it into their L1 for managing ongoing communication. However, participants reported that translation, which appeared at an earlier L2 acquisition stage, disappeared after a few years (most participants said after three or four years) and by the time it faded they were able to understand sentences naturally as they heard the language. For instance, most intermediate bilinguals reported that they still translated L2 into L1 when they needed, while high bilinguals reported that they did not need to use their L1 to understand L2. With improved L2 proficiency, they did not need to mentally repeat sentences to process them. However, their tendency for remembering and holding information did not totally disappear; it appeared in different forms. This is discussed under the theme ‘Effort is continuous’.

#### 5.2.2. Effort is Continuous

As seen in the first theme, while bilinguals in the present study were practicing bilingualism they developed strategies to compensate for their lack of L2 language skills. Their continuous effort in using strategies was observed during the interview. Throughout their journey of mastering L2, they employed different strategies based on their level of L2 proficiency. This change will be discussed under three categories: stabilized L2 system, improving L2, and degree of monitoring. 

Stabilized L2 system. All participants mentioned that they felt comfortable using English as their years spent in the United States increased. For example, participant F mentioned how her strategy had changed as her L2 improved: “*I feel more confident in speaking in English, less translation.*” She indicated that she was less likely to translate what she heard as her L2 proficiency improved. The bilinguals’ needs for translating from L2 to L1 or remembering sentences for the purpose of understanding seemed to decrease as they felt confident in their L2. The high bilinguals also indicated that their stabilized L2 helped them to use their L2 with less effort. Participant D mentioned: “*I got comfortable using English; in a lot of cases, I understand English as [well as] I understand Korean.*” He indicated that his L2 system had stabilized to the degree that he was able to process English as he processed Korean. Bilinguals’ developed L2 seemed to free them from using some of the L2 strategies that they built up while acquiring their L2.Improving L2. Even though high bilinguals reported that they no longer needed to hold what they heard in L2 in mind for translation, they tended to remember sentences from conversations for a different purpose. Participant A reported his habit of using English:
*When I talk to my friends, sometimes I hear expressions that I want to use but I haven’t used so I repeat the sentences in my mind and try to memorize them, because I want to use them later if I have any chances to use them*.During conversations with native speakers of English, he caught expressions that he did not use in English and then he tried to remember them for future use. The interview data showed that bilinguals were attentive to L2 most of the time when they used the language. Similar to their earlier L2 stage, bilinguals employed remembering strategies continuously but the goal of this practice was not the overcoming of their lower L2 proficiency but improving their L2 skills.Degree of monitoring. The two bilingual groups also showed different degrees of monitoring. For example, in a group interview, the intermediate bilinguals agreed that they tended to be attentive to their L2 whenever they used it, while the high bilinguals agreed that they often did not notice their attention to the L2. Following is an interview with the high bilingual group about their remembering strategy:
D:*No, I don’t, if I am able to understand it, I understand it right away, if I don’t understand, I ask right away what the person tries to say, but I don’t try to remember information. When I first learned English, I mean when I tried to learn English, I did, that was how I learned English, but I do not do it any more I guess, I just ask if I missed something*.
Interviewer:*What about you?* 
B:*My case is the same, when I started trying to learn English, I tried to remember as much as possible, but I don’t know whether I do that or not. Hmm, I do that sometimes when someone use words that I haven’t used, new expressions, yeah, then I would remember it so I can use it, but not often, I feel quite comfortable with English, (I have) been living here about eight years*.
Interviewer:*Do you guys agree on this?* 
C:*Yeah, English is quite comfortable, sometimes, more comfortable than Korean, [I] do not need to think that much when I use English, I guess*.D reported that he had tried to remember words, expressions, and sentences as he heard them when he started to learn L2, but he “*does not do it anymore*.” This pattern was also observed by all the high bilingual participants in their individual interviews. They were not sure whether they consciously remembered the information anymore; as B said, “*I don’t know whether I do that or not*.” In addition, they focused on and remembered less of their L2 output than they used to. C pointed out that she was quite comfortable with English, so she did not need to focus on her language. However, bilinguals in this study were still attentive to what they heard; as B said: “*I do that sometimes when someone use words that I haven’t used*.” Her comment shows that she noticed an expression that was not familiar with. This indicates that she was still attentive to her L2 input even though she did not realize until she heard a new expression. B felt “*quite comfortable with English*” after living in the United States “*for about eight years*,” so she did not need to use strategies as intensively as she did when she was in an earlier stage of L2 learning. This suggests that she established a stabilized L2 language system during her stay in the United States, so she was able to use her L2 without using strategies she developed when she first came to the United States. The interview data showed that bilinguals used remembering and monitoring strategies in order to compensate for their lack of L2 proficiency more in the earlier stages of L2 acquisition. Their longtime practice of using these strategies developed their tendency of remembering and monitoring their L2. Even though the high bilinguals did not need to use their strategies as intensively as intermediate bilinguals did, supposedly due to their stabilized L2 system, they still tended to remember what they heard when they encountered new expressions within their interests, and they kept improving their L2 by using these strategies.

## 6. Conclusions and Discussion

The purpose of the study was to examine a possible WM difference among three language groups: Korean near-monolingual, Korean–English intermediate, and Korean–English high bilingual groups, and to explore bilinguals’ possible cognitive changes in relation to their everyday language practices. 

The DSTs were used to examine WM capacity in the three language groups. It is important to note that the IQs of the three language groups were similar to each other; therefore, it is unlikely that the participants’ IQs influenced the differences in DSTs. The MANOVA analysis of the tests revealed statistically significant differences in DSTs among the three different language groups and the follow-up post hoc analysis showed where the difference lay. The intermediate bilingual group outperformed other language groups at remembering the order of the digits presented both in visual and auditory numeric forms. They also scored higher when recalling the reversed order of digits presented in the auditory form. This suggests that the intermediate bilinguals have higher visual/auditory memory maintenance as well as higher auditory memory manipulation capacity than the two other language groups in the study. The interesting point is that intermediate bilinguals performed better than the other language groups, with large effect sizes, especially in both the auditory forward and the backward DSTs. This finding indicates that their bilingual experience somehow significantly develops their auditory WM memory capacity. Meanwhile, the high bilingual group performed similar to the near-monolingual group, suggesting that this WM advantage only appeared in the intermediate bilingual group. The findings of DSTs provide evidence that the intermediate bilinguals’ relatively shorter time of L2 practice and exposure might grant an advantage of their WM. However, the results fail to support high bilinguals’ WM advantage over monolinguals or intermediate bilinguals. This is not consistent with previous findings on bilinguals’ advantages [7,33], and supports neither of the WM hypotheses. 

However, the interview data suggest a possible explanation for the results of DSTs. According to the analysis of the interview, most bilinguals developed their own strategies while learning their L2. Even though the strategies varied between individuals, there were common patterns in their strategies: monitoring, holding and replaying L2 information for understanding. Especially, in the earlier stage of L2 learning (often reported by intermediate bilinguals), bilinguals paid attention to what they were hearing so they could hold that information. This information was retained and replayed in their mind in case they did not understand the meaning as they heard. As they acquired an ability to understand the L2 input in the moment that they heard, mostly in the case of the high bilinguals, the frequency of using memorizing and replaying strategies decreased to the point where they could not even recognize that they were using them. The high bilinguals used their strategies for other purposes: to remember new words, sentences, or expressions that they wanted to use in the near future, or to monitor the accuracy of their L2 output. 

The interview data overall showed that both bilingual groups used strategies of monitoring and holding L2 input but the intermediate bilinguals used the strategies of memorizing and replaying more intensively than the high bilinguals. This indicates that dual language practices might impose an extra cognitive load since bilinguals need to hold their L2 information to process it. According to the cognitive enrichment hypothesis [65,66], continuous practice of life-long activities enhances cognitive functions that are related to those practices [67,68,69]. Under this hypothesis, it is possible that the extra cognitive load might force bilinguals to manage the extra load and somehow enhances their WM. A cross-examination of findings from both the test results and the interview data shows that the intermediate bilinguals reported intensive use of remembering strategies and they scored higher in the auditory DSTs. Therefore, it can be interpreted that the intermediate bilinguals’ continuous practices of monitoring, remembering and replaying strategies served as mental training and enhanced their performance on DSTs. Furthermore, intermediate bilinguals’ description of remembering and replaying practices was mainly about remembering auditory information. This might explain why they showed stronger performance, particularly in the auditory forward/backward DSTs. Their auditory WM might be developed while managing continuous cognitive demands that are imposed due to their lack of L2 proficiency.

However, the high bilingual group, unlike the intermediate bilingual group, did not perform better on DSTs than the monolingual group, while the interview data showed that they also used the strategies in the process of learning a new language. Why did they not experience the same benefits that intermediate bilinguals enjoyed? In interview data, it appeared that dual language practices became natural for the high bilinguals. In the beginning stages of acquiring their L2, they used remembering strategies to understand L2, as the intermediate bilinguals did. However, once their language reached a level where they processed the L2 input as they heard it, they did not need to retain and replay the L2 input. Instead, they tended to remember new words and expressions whenever they encountered them for future use and the frequency of facing new words decreased as their L2 proficiency improved. The average years of residency in the USA were 8.25 for the high bilinguals. As their L2 proficiency got higher during their stays in the United States, they became fairly comfortable with English and their cognitive demands for remembering L2 decreased. Therefore, it can be supposed that high bilinguals’ degree of mental demands from using strategies, particularly strategies related to holding and replaying L2 information, was relatively lower than that of the intermediate bilinguals, so that it did not grant a WM advantage for the high bilingual group while it did for the intermediate bilingual group.

These findings are in line with the results of current studies, which suggest that bilinguals’ cognitive advantages might result from using specific cognitive skills that bilingualism requires. For example, studies suggest that bilinguals who frequently switch between languages show an advantage in cognitive control over bilinguals who do not switch between languages [70,71]. Macnamara and Conway [72] also show evidence that bilinguals’ enhanced cognitive control and WM depend on the degree of cognitive demands that they recruit for managing two languages. The present study shows that the intermediate bilinguals with high demands in remembering auditory information scored significantly higher in auditory DSTs than the two other groups, while high bilinguals with lower demands showed similar digit spans to the near-monolinguals. This finding adds evidence to the body of studies that being bilingual does not guarantee WM advantage but bilinguals’ advantages might be related to the cognitive training they are engaged in while using two languages. 

In sum, these overall findings partially support the notion that dual language use serves as mental training; the WM of the intermediate bilinguals was enhanced, while that of high bilinguals was not. The study suggests that the intermediate bilinguals might develop their WM because of the high WM demands that bilingualism imposes. While using two languages, they needed to continuously monitor, memorize and replay what they are hearing in order to overcome their lack of language proficiency. Meanwhile, the high bilinguals might not enjoy any advantage since their higher L2 proficiency allows them to process incoming L2 information instantly so that their cognitive demands for processing L2 are insufficient to enhance their WM capacity. The findings of the study overall suggest that knowing two languages does not guarantee bilinguals’ WM advantage over monolinguals. Their advantage might depend on their unique L2 environment, where bilinguals need to use certain cognitive functions and also depend to a certain degree on the cognitive skills that bilinguals recruit for managing two languages. This study also suggests that bilinguals’ cognitive advantage can be population-specific. Therefore, how bilinguals manage two languages and what language environment they are situated in needs to be considered when their cognitive advantages are being investigated.

## 7. Limitations and Recommendations for Future Research

Even though the study provides insightful findings, there are a few limitations that need to be addressed as guidance for future studies. First of all, the scope of the study is limited due to the small sample size. In order to generalize the findings to a Korean–English bilingual population, further studies with larger sample sizes are required. 

Secondly, the study could be extended to longitudinal studies. In this study, the high bilingual group did not show a WM advantage even though they reported that they had used their remembering strategies in their earlier L2 learning stages. With the cross-sectional design employed for the study, it is impossible to determine whether the high bilinguals might have enjoyed a WM advantage at an earlier L2 learning stage, then lost it as their cognitive demands decreased, or whether they might not have had a chance to enhance their WM as they developed their L2 skills before they enjoyed their WM advantage. The longitudinal designs might answer these questions and provide a clearer picture of the relationship between bilinguals’ continued language practices and cognitive enhancement. 

Thirdly, the implication of findings is limited to Korean–English bilinguals. The findings provide evidence that bilinguals’ WM advantage might result from their extensive remembering strategies. However, it is possible that they used the remembering strategies extensively because Korean word order (Subject + Object + Verb) is very different from English word order (Subject + Verb + Object ) so that they hold the information while translating their L2 to L1. Meanwhile, it is also possible that they needed to hold the information to process their L2 due to a lack of L2 proficiency regardless of the word order difference between the two languages. In order to understand clearly where the advantage comes, further studies need to be conducted with different populations. Replicating this study with bilinguals who speak two similar languages (e.g., Spanish–English bilinguals or French–English bilinguals) might show whether WM advantages result from holding L2 information in mind for processing. Comparing intermediate bilinguals of Korean and English with intermediate bilinguals of Spanish and English might reveal whether the difference in sentence structure potentially demands heavier memory resources and eventually affects bilinguals’ WM.

Lastly, recent studies suggest that certain Asian cultures offer environments where behavioral control is emphasized and that the behavior control might affect their executive control [73,74,75]. In the present study, the high bilinguals’ relatively longer stays in the United States might weaken ties to Korean culture, which might have an influence on the results of the study. Therefore, future research may need to control for a possible cultural effect on bilinguals’ cognitive changes when bilinguals experience distinctively different cultures.

## Figures and Tables

**Figure 1 brainsci-07-00086-f001:**
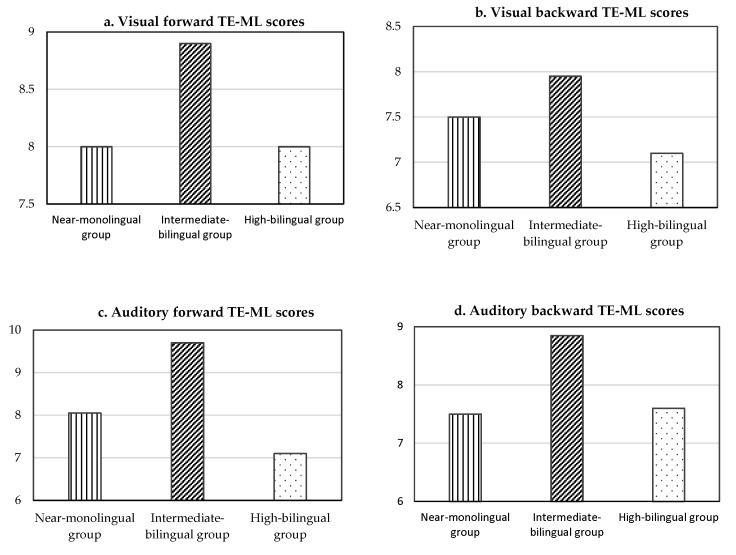
Group mean comparisons in TE-ML scores. (*a*) Visual forward two-error maximum length of three language groups; (*b*) Visual backward two-error maximum length of three language groups; (*c*) Auditory forward two-error maximum length of three language groups; (*d*) Auditory backward two-error maximum length of three language groups.

**Table 1 brainsci-07-00086-t001:** TOEFL iBT score scales.

Section	Range of Scores	What It Means
Reading	0–30	0–14 (low)
		15–21 (intermediate)
		22–30 (high)
Listening	0–30	0–14 (low)
		15–21 (intermediate)
		22–30 (high)
Speaking	0–4 points, converted into a 0–30 scale	0–9 (weak)
		10–17 (limited)
		18–25 (fair)
		26–30 (good)
Writing	0–5 points, converted into a 0–30 scale	1–16 (limited)
		17–23 (fair)
		24–30 (good)

Note: Retrieved from [50].

**Table 2 brainsci-07-00086-t002:** The description of participants.

	SES		Education		Gender		Daily L2 Exposure (Hour)	Year of Stay in L2 Speaking Country
	No. of M.C.	No. of U.M.C.	No. of U.G.S	No. of G.S	M	F		
Near-Monolingual (*N* = 20)	15	5	14	6	9	11	1.88	Less than 2 months
Intermediate Bilinguals (*N* = 20)	16	4	13	7	10	10	8.9	4.14
High Bilinguals (*N* = 20)	18	2	13	7	11	9	8.8	8.25

Note: Intermediate Bilinguals: Bilinguals with intermediate L2 proficiency; High bilinguals: Bilinguals with high L2 proficiency; SES: Socioeconomic status; M.C.: Middle class; U.M.C: Upper-middle class; U.G.S.: Undergraduate students; G.S.: Graduate students; M: Male; F: Female.

**Table 3 brainsci-07-00086-t003:** Descriptive statistics of age and Cattell scores of three language groups.

	Language Groups		
	Monolingual (*N* = 20)	Intermediate Bilinguals (*N* = 20)	High Bilinguals (*N* = 20)
Age	24.50 (4.01)	24.45 (4.07)	23.50 (2.373)
Cattell	59.35 (5.65)	59.15 (5.35)	58.90 (5.34)

Note: Cattell = Cattell Culture Fair Intelligence Test. Standard Deviations appear in parentheses next to means. Intermediate bilinguals: bilinguals with intermediate L2 proficiency.

**Table 4 brainsci-07-00086-t004:** Participants’ background information.

Participants	Language Group	Age	Gender	TOEFL Score	Years in USA	Field of Study
A	H.B	22	Male	102	8	Pharmacy
B	H.B	25	Female	97	7.2	Management
C	H.B	24	Female	98	6	Business
D	H.B	27	Male	94	10	Physical therapy
E	I.B	22	Female	70	4.5	Management
F	I.B	25	Female	86	4.8	Biology
G	I.B	28	Male	88	3.2	Education
H	I.B	29	Male	80	4	Computer science

Note: H.B: Korean–English bilingual group with high L2 proficiency; I.B: Korean–English bilingual group with intermediate L2 proficiency.

**Table 5 brainsci-07-00086-t005:** Themes and categories used in the study.

Theme	Categories
Strategies in using 2nd language	1. Difference between L1 and L2
	2. Monitoring languages
	3. Holding information
Efforts is continuous	1. Stabilized L2 system
	2. Improving L2
	3. Degree of monitoring

**Table 6 brainsci-07-00086-t006:** Correlation between dependent variables from digit span tasks.

	1	2	3	4	5	6	7	8	9	10	11	12	13	14	15	16
1. VF TE-ML	1															
2. VF TE-TT	0.850 **	1														
3. VF ML	0.740 **	0.551 **	1													
4. VF MS	0.741 **	0.423 **	0.912 **	1												
5. VB TE-ML	0.535 **	0.399 **	0.574 **	0.544 **	1											
6. VB TE-TT	0.399 **	0.360 **	0.360 **	0.320 *	0.808 **	1										
7. VB ML	0.525 **	0.328 *	0.629 **	0.643 **	0.745 **	0.534 **	1									
8. VB MS	0.502 **	0.341 **	0.574 **	0.610 **	0.732 **	0.510 **	0.900 **	1								
9. AF TE-ML	0.389 **	0.358 **	0.437 **	0.389 **	0.472 **	0.324 *	0.500 **	0.534 **	1							
10. AF TE-TT	0.298 *	0.295 *	0.329 *	0.270 *	0.319 *	0.211	0.364 **	0.361 **	0.919 **	1						
11. AF ML	0.501 **	0.505 **	0.521 **	0.482 **	0.591 **	0.325 *	0.530 **	0.605 **	0.694 **	0.539 **	1					
12. AF MS	0.509 **	0.487 **	0.544 **	0.527 **	0.602 **	0.359 **	0.573 **	0.655 **	0.731 **	0.496 **	0.926 **	1				
13. AB TE-ML	0.365 **	0.306 *	0.385 **	0.382 **	0.503 **	0.361 **	0.527 **	0.595 **	0.701 **	0.574 **	0.551 **	0.623 **	1			
14. AB TE-TT	0.287 *	0.222	0.239	0.247	0.459 **	0.339 **	0.479 **	0.558 **	0.557 **	0.455 **	0.381 **	0.450 **	0.896 **	1		
15. AB ML	0.447 **	0.379 **	0.534 **	0.516 **	0.604 **	0.416 **	0.544 **	0.604 **	0.668 **	0.532 **	0.724 **	0.758 **	0.715 **	0.515 **	1	
16. AB MS	0.432 **	0.359 **	0.524 **	0.520 **	0.535 **	0.355 **	0.507 **	0.575 **	0.700 **	0.548 **	0.713 **	0.759 **	0.740 **	0.493 **	0.958 **	1

VF: Visual forward, VB: Visual backward, AF: Auditory forward, AB: Auditory backward, TE-ML: Two-Error Maximum Length, TE-TT: Two-Error Total Trial, ML: Maximal Length, MS: Mean Digit Span. **: Correlation is significant at the 0.01 level (two-tailed); *: Correlation is significant at the 0.05 level (two-tailed).

**Table 7 brainsci-07-00086-t007:** Correlations between selected WM variables.

	1	2	3	4	M	SD
1. Visual forward TE-ML	1				8.30	1.38
2. Visual backward TE-ML	0.535 **	1			7.52	1.19
3. Auditory forward TE-ML	0.389 **	0.472 **	1		8.68	1.57
4. Auditory backward TE-ML	0.365 **	0.503 **	0.701 **	1	7.98	1.69

TE-ML: Two-Error Maximum Length.**: Correlation is significant at the 0.01 level (two-tailed).

**Table 8 brainsci-07-00086-t008:** One-way ANOVAs with two-error maximum lengths as dependent variables and language groups as independent variables.

	Levene’s		ANOVAs		
	*F* (2, 57)	*p*	*F* (2, 57)	*p*	*η* ^2^
Visual forward TE-ML	1.18	0.315	3.02	0.056	0.096
Visual backward VB TE-ML	0.34	0.716	2.72	0.074	0.087
Auditory forward AF TE-ML	0.34	0.711	4.41	0.017	0.134
Auditory backward AB TE-ML	0.98	0.380	7.95	0.001	0.218

Note: TE-ML: two-error maximum length.

**Table 9 brainsci-07-00086-t009:** Mean differences in two-error maximum lengths between language groups expressed as Cohen’s *d.*

	N. M.		I. B.		H. B		N. M. vs. I. B. (Cohen’s *d*)	N. M. vs. H.B. (Cohen’s *d*)	I. B. vs. H. B. (Cohen’s *d*)
	M	SD	M	SD	M	SD			
VF TE-ML	8.00	1.52	8.90	1.73	8.00	1.08	0.90 * (0.62)	0.00	0.90 * (0.73)
VB TE-ML	7.50	1.28	7.95	1.10	7.10	1.07	0.45	0.40	0.85 * (0.78)
AF TE-ML	8.05	1.82	9.70	1.42	8.30	1.54	1.65 * (1.01)	0.25	1.40 * (0.95)
AB TE-ML	7.50	1.43	8.85	1.17	7.60	1.59	1.35 * (0.94)	0.10	1.25 * (0.83)

Note: VF: Visual forward, VB: Visual backward, AF: Auditory forward, AB: Auditory backward, TE-ML: Two-Error Maximum Length, N. M.: Near-monolingual group, I. B.: Intermediate bilingual group, H. B.: High bilingual group. Cohen’s *d* is reported when it is significant. *: The mean difference is significant at the 0.05 level.

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
