# Peer review of "Bilinguals’ Working Memory (WM) Advantage and Their Dual Language Practices"

_brainsci, 2017, doi:10.3390/brainsci7070086_

Round 1

Reviewer 1 Report

Title: Bilinguals working memory advantage and dual language practices

Review: Brain Sciences

The purpose of this study was to compare working memory differences or similarities between monolingual’s and bilinguals. The sampling included adult Korean near monolingual group and two young adult Korean-English Korean with different levels of language proficiency. The studies sought to investigate how bilingualism influences memory use. The study employed a computerized digit span task well represent the numbers visually and auditorily. The instrument also allowed for measuring forwarded span and backward digit span. The study also included semi structured interviews among voluntary participants. A one-way MANOVA was used to compare the three different language groups (near monolingual’s, intermediate bilinguals, and high bilinguals) which yielded a significant group effect. Interestingly, there are no significant differences between near monolingual and high bilingual groups but there was is difference between near monolingual groups and intermediate groups. That is the results suggest that the intermediate bilinguals perform better than the other two language groups while the high bilingual groups did not differ from the near bilingual group. The authors concluded that bilingualism might have a different effect on working memory, depending on  L2 proficiency.

This study appears to be a competently carried out. The introduction covers the major issues within the literature and the paper as a whole is clearly written. Although the statistical analysis was appropriate for giving us a sense of the intercorrelation among the variables, the outcomes do not appear to make sense. I’m not sure why intermediate bilinguals would do better than the other two groups. Although the authors took time to make sure that age and intelligence did not separate the three groups, I’m not sure the TOEFL scores were sensitive enough related to gradations of bilingualism. The fact that the intermediate group did slightly better than the other group may reflect a number of possibilities which cannot be determined from the study. In addition, there has been some controversy as to whether the digit span task merely taps some sort of phonological processing skills rather than working memory. Rosen and Engle (2007), Colom and colleagues (2005) suggest a phonological code is involved in both forward and backward serial recall. This suggests that the backward span and the forward span are merely a measure of phonological short-term memory rather than tapping the executive processing component of working memory. Regardless, my major concern is that we need a more thorough explanation as to why the intermediate group excelled on a number of the digit span measures.

Author Response

Thank you so much for your efforts in reviewing the paper. Please find my responses to your comments in the attached file. I hope the revised paper will meet your expectations.

Reviewer 2 Report

The authors compare three groups—Korean near monolinguals, intermediate Korean-English bilinguals, and advanced Korean-English bilinguals—on a combined short-term memory/working memory task. Participants completed both auditory and visual versions of forward and backward digit spans. Across all measures, the intermediate bilinguals outperformed the near monolinguals and the advanced bilinguals. The near monolinguals and the advanced bilinguals did not differ from one another. The authors suggest that the cognitive demands required to manage two languages before these are automatic processes cause the enhancement.

This is an interesting study that I think should be published. Overall it is well written and it is about a controversial topic where more information is needed. I have some general suggestions as well as some the authors might want to consider to increase how convincing the results are.

Comments and suggestions:

Remove last sentence of introduction. It makes it read like a second abstract.

I find the multiple outcome measures for each test strange.  The correlations are around .75, which doesn't reach the standard for multicollinearity, but conceptually there is no explanation for why both mean length and maximum length are equal DVs. It's redundant. The authors should either explain why both are equally important and informative, or choose the more informative measure and present the results with one outcome per task.

I like the choice of a MANOVA for the analyses for the multiple test scores. However, the authors state that part of the reason for conducting an ANOVA is to avoid the problems associated with post-hoc tests, but then proceed to conduct multiple post-hoc tests. Also, that's a problem with MANOVAs, if one actually wants to know how groups differ (on which outcome variables) then a MANOVA is not appropriate and a series of corrected ANOVAs are more appropriate. To this end, I disagree with the use of discriminant function analysis as a follow up test. DFA is mathematically a MANOVA in reverse. It provides some additional information, but it's not following up, it's providing the same information from a different perspective. Again, if it's important where the differences are, the statistics test should reflect that in the first place. The follow up ANOVA with a canonical DV is also redundant with the MANOVA. I suggest conducting the MANOVA and reporting effect sizes along with the means and standard deviations with each of the measures for the 3 groups. This may be better presented in a figure than a table. Further, given that the authors are using a MANOVA over a single composite score, they should consider discussing their findings in terms of an advantage of several related memory constructs (not those words per se) rather than referring to it as a single construct.

The authors say that 7 variables were entered into the MANOVA, but 8 were. The authors may choose to enter 4.

The authors state, "This finding invites two possible explanations. One is that high-bilinguals might not have any chance to enhance their WM as they developed their L2 skills before they enjoy their WM advantage." This explanation is not clear. Why might this be the case?

The Discussion is lacking. There needs to be more about how these findings make sense and there needs to be more about the limitations and direction for future research. The authors suggest that the intermediate bilinguals are doing the most cognitive work and thus, are seeing an advantage. First, remind the readers that the groups were equivalent on IQ, so it's not just a sampling error fluke that the intermediate group has higher cognitive functioning all around. Second, point to supporting literature. For example, the two references below examine cognitive functions among simultaneous interpreters with the premise that they are bilinguals having to use "extreme language control" and have "high bilingual management demands," respectively. These papers seem like they would support your findings: the more cognitively demanding being bilingual is, the more an advantage should appear (and potentially only appear under those circumstances.) Talk about how this could explain discrepancies on the findings of a WM advantage for bilinguals that you set up in the paper. Might additional bilingual management demands of the not-yet L2 automated account for some of these discrepancies depending on the sample used in these other papers? (There could be sampling error pushing results in both directions as well.) Third, discuss limitations and future directions. While you attempted to control as much as possible, 20 is not a large sample size statistically (not bad for this type of bilingual research though.) Do you think the findings were especially strong because of the language pair? In other words, if both languages were subject-verb-object would the results be the same? Future directions might include 1) a longitudinal study, 2) replicating this study with another language pair, for example English and another language where the verb falls at the end of the sentence, or 3) comparing intermediate bilinguals of Korean and English with intermediate bilinguals of Spanish and English or another pair where the sentence structures are more similar and potentially demand fewer memory resources.

Hervais-Adelman, A., Moser-Mercer, B., Michel, C. M., & Golestani, N. (2015). fMRI of simultaneous interpretation reveals the neural basis of extreme language control. Cerebral Cortex25(12), 4727-4739.

Macnamara, B. N., & Conway, A. R. (2014). Novel evidence in support of the bilingual advantage: Influences of task demands and experience on cognitive control and working memory. Psychonomic Bulletin & Review21(2), 520-525.

In sum, I think the paper offers a substantial contribution, but the authors are not giving their research enough credit and are not writing it in a way that would be convincing to others. If they can make the Discussion more substantial this will be an excellent contribution.

Author Response

(The authors gave the same response as above.)

Reviewer 3 Report

p.p1 {margin: 0.0px 0.0px 0.0px 0.0px; font: 11.0px Calibri} p.p2 {margin: 0.0px 0.0px 0.0px 48.0px; text-indent: -48.0px; font: 11.0px Calibri} span.s1 {font-kerning: none}

The submitted manuscript tested for a bilingual advantage in working memory. To this end, tests of number recall were given to Korean monolinguals, moderate proficiency Korean-English bilinguals, and high proficiency Korean-English bilinguals. Data from the recall tests were supplemented with interview data. Results indicated a bilingual advantage for moderate proficiency bilinguals but not high proficiency bilinguals. This finding was explained by invoking the interview data, which suggested that moderate proficiency bilinguals might engage in more working memory demanding strategies, thereby enhancing their working memory. 

I think the article contributes to the literature on bilingual advantages, though I do have some concerns. I’ll list my larger concerns first, followed by section-by-section comments.

- While I found the study interesting and informative, I am concerned about the novelty of the study (a factor that this journal seems to value, based on the evaluation mechanism). As I mention below, there have been 27 studies on bilingual advantages (or lack thereof) in working memory. There is, however, some novelty (i.e., testing different levels of proficiency and including interview data), which may be sufficient.

- A main advantage of the study is that two different levels of proficiency were tested. This factor gives the study novelty and produces the most interesting results (i.e., that moderate proficiency bilinguals have an advantage but high proficiency bilinguals do not). However, it seems likely that proficiency is confounded with age of acquisition. In other words, it is likely that moderate proficiency was correlated with late age of acquisition. Potentially, it is age of acquisition that is important here, rather than proficiency. This confound weakens the study in my view.

- Unlike most studies that test for a bilingual advantage, a linguistic-based task was used in the current study. What is the rationale for using a linguistic-based task, rather than a non-linguistic task, such as the frog matrices test (Morales, Calvo, & Bialystok, 2013)? The use of a linguistic-based task leads to an impure measure of working memory in bilinguals in my view.

Introduction

- Based on my reading of the literature, some of the claims in the Introduction, which I list below, seem unsupported. 

- “the consistent evidence of cognitive advantages in inhibitory control and cognitive flexibility” - the work by Paap and many others suggest that the evidence is not consistent, with many studies failing to find a bilingual advantage in inhibitory control and cognitive flexibility.

- “the recently accepted view on EF’s” and “widely accepted model” (in reference to Miyake’s 3-part model). This model is not universally accepted in my view. In fact, Bialystok, who has done much of the seminal work on bilingual advantages, has recently expressed criticisms of this model (see Bialystok 2017 in Psychological Bulletin).

- “not many studies have been conducted on bilinguals’ WM” - it depends on what is meant by “not many” but a recent meta-analysis by Grundy & Timmer (2016) found 27 studies on bilinguals’ WM, which to me seems more than “not many.”

Methods

- The monolinguals live in South Korea and the bilinguals live in the U.S., so culture seems to be confounded with language status. This may be problematic given some evidence that Korean culture is associated with enhanced executive functioning (Yang, Yang, & Lust, 2011).

- For analyses of IQ and age, why was alpha set at .025, as opposed to the more common .05?

Results

- The analyses (namely, the use of MANOVAs and centroids) are unconventional for this kind of study, though they seem reasonable.

- While I really appreciate the interview data, they seem disconnected with the rest of the experiment. I think they need to be better integrated.

Discussion

- The enhanced performance by the moderate proficiency bilinguals (but not the high proficiency bilinguals) is surprising but reasonably well explained. However, more discussion of how this finding fits with previous research is needed. It seems that it doesn’t fit very well, given that increased proficiency is usually correlated with stronger cognitive advantages (see for example, Luk, De Sa, & Bialystok, 2011).

Author Response

(The authors gave the same response as above.)

Round 2

Reviewer 1 Report

This my second review of the manuscript. My primary concerns earlier were  that there was not an adequate explanation as to (1) why intermediate bilinguals would do better than the other two groups, (2) whether the TOEFL scores were sensitive enough related to gradations of bilingualism and (3) whether the digit span task merely taps some sort of phonological processing skills rather than working memory. I feel the authors responded where possible to my concerns. However, their explanation was more focused on providing detail on the analysis procedures and not the “Why” ---especially to my first concern. I think the authors did the best they could with the data, but I’m wondering if they may consider the inverted “U” findings in the cognitive effort literature. I’m not requiring this but to argue the strategies were better for intermediate bilinguals is more post-hoc than theoretically explaining the mechanism that underlie bilingualism.

Author Response

Thank you so much for reviewing the manuscript. Your comments were very helpful to improve the quality of the manuscript. I really appreciate your time and effort.

Reviewer 3 Report

The author did a considerable amount of work on the revision, for which I am appreciative. I think the manuscript has been significantly improved, though a few of my concerns remain unaddressed.

The concern that age of acquisition (AoA) is confounded with proficiency remains. In the author’s reply, they state that indeed there is likely a correlation between age of acquisition and proficiency in the current study. This is concerning to me, given that age of acquisition has been associated with executive control (see, for example, Soveri, Rodriguez-Fornells, & Lain, 2011; Tao, Marzecová, Taft, Asanowicz, & Wodniecka, 2011). In my view, this needs to be addressed both in the Participants section and the Discussion section. In the Participants section, more information on participants’ AoA needs to be provided. In the Discussion section, this factor needs to be acknowledged as an uncontrolled factor that could have influenced the results.

The concern that culture is confounded with proficiency also remains. There is evidence that higher proficiency in a language correlates with stronger affiliation with the language’s corresponding culture (see relevant work by Clement and Noels, among others). If the moderate proficiency group has weaker ties to US culture (but stronger ties to Korean culture), and if Korean culture is associated with better executive control, then that could explain the current results. In my view, this needs to be discussed in the Discussion.

This wording would still not be accepted by many researchers in the field: “While it is well established that bilinguals have cognitive advantages in inhibitory control and cognitive flexibility…” Researchers, such as Paap, Klein, Morton, De Bruin, would not agree that this finding is well-established.

Author Response

(The authors gave the same response as above.)
